# Quantitation of Silica Contents in Lung Explants of Transplanted Patients: Artificial Stone-Induced Silicosis vs. Idiopathic Pulmonary Fibrosis

**DOI:** 10.3390/ijerph18147237

**Published:** 2021-07-06

**Authors:** Elizabeth Fireman, Abed Elrahman Mahamed, Dror Rosengarten, Noa Noach Ophir, Mordechai Reuven Kramer

**Affiliations:** 1Laboratory for Pulmonary and Allergic Diseases, Tel-Aviv Medical Centre, Tel-Aviv 6423906, Israel; abed.alrhman7@gmail.com (A.E.M.); noano@post.tau.ac.il (N.N.O.); 2Environmental and Occupational Medicine, Sackler Faculty of Medicine, Tel-Aviv University, Tel-Aviv 6997801, Israel; 3Institute of Pulmonary Diseases, Sackler Faculty of Medicine, Tel-Aviv University, Petah Tekvah 4947492, Israel; drorroz@gmail.com (D.R.); Kremerm@clalit.org.il (M.R.K.); 4Rabin Medical Centre, Sackler Faculty of Medicine, Tel-Aviv University, Petah Tekvah 4947492, Israel

**Keywords:** artificial stone-induced silicosis, X-ray fluorescence, quantitation of silica

## Abstract

Spectrophotometric techniques provide qualitative but not quantitative data on lung particles. We aimed to quantitate silica content in biopsies of lung-transplanted silicosis patients by applying X-ray fluorescence (XRF) spectrometry. Lung biopsies of 17 lung-transplanted artificial patients were quantitated for silica and other minerals particles by Niton XL3 XRF spectrometry. Occupational and clinical history data were assessed. Lung biopsies of artificial stone-induced silicosis (ASIS) patients contained significantly higher levels of silica compared to those of idiopathic pulmonary fibrosis (IPF) patients (7284.29 ± 4693.75 ppm vs. 898.88 ± 365.66 ppm, *p <* 0.0001). Silica content correlated negatively with age, body mass index, and pulmonary function test results. A 1128 ppm silica cut-off value yielded 100% sensitivity and 94% specificity for predicting ASIS (AUC = 0.94, *p <* 0.0001). In conclusion, XRF measurements in lung biopsies can differentiate between silica and mineral particles in ASIS and IPF.

## 1. Introduction

Silicosis is a primary pneumoconiosis involving fibronodular lung disease caused by inhalation of silica dust. Quartz, the most common form of crystalline silica, is abundantly present in granite, slate, and sandstone [1,2]. The marble industry in Israel, which manufactures kitchen and bathroom countertops, is based mainly on artificial stone that contains more than 90% silica (SiO_2_) as a substitute for marble or granite. Overexposure of these marble workers to high concentrations of silica dust in their workplace may lead to silicosis, a chronic granulomatous disease characterized by inflammatory processes followed by fibrosis of lung tissue. The outbreak of this new type of silicosis was reported in Israel in 2012 [3], and 32 patients with silicosis have since been transplanted in Rabin Medical Center, the only lung transplantation center in Israel. This number of silicosis patients reflects the magnitude of the epidemic in Israel. Other cases were also found in Spain and Italy, followed by Australia and the USA [4,5,6,7,8].

The diagnosis of pneumoconiosis may be suggested by occupation, history, chest radiography, and routine light microscopy, but a multidisciplinary approach that includes electron microscopy plus analytical and quantification techniques is usually necessary to identify inhaled particles and for studying particle localization and exposure levels. Scanning electron microscopy (SEM) with secondary electron imaging has been widely used in the study of pneumoconiosis since it allows topologic examination of large fields at magnifications of 20–100,000 X, with identification and shape determination of particles too small to be seen by light microscopy [9].

In our previous work we used SEM to redefine idiopathic interstitial lung disease into occupational lung diseases by analysing the chemical composition of inhaled dust particles in induced sputum and/or lung biopsy specimens [10]. As that study was limited to qualitative analysis, we subsequently applied an X-ray fluorescence (XRF) device to describe quantitative levels of minerals present in induced sputum samples of workers exposed to artificial stone dust [11]. Quantitation of metals in other biological samples such as endothelial cells [12], atherosclerotic plaques [13] and dental tissues [14] by XRF were also shown. Mineral detection by XRF and ICP-MS show a good correlation between both methods [15].In this context we expanded the scope of the analyses and evaluated, the biopsies of patients who had undergone lung transplantation due to artificial stone-induced silicosis (ASIS). We compared the findings with those of biopsies retrieved from patients who had undergone lung transplantation due to idiopathic pulmonary fibrosis (IPF).

## 2. Materials and Methods

The study was approved by the Institutional Review Board Local Committee of the Tel Aviv Medical Center (0435-14-TLV) and Rabin Medical Center (0541-14) and patient consent for this retrospective analysis of medical records was waived. We conducted a retrospective review of the medical records of 17 ASIS patients (mean age 48.7 ± 8.8 years and 17 IPF patients (mean age 58.2 ± 6.3 years) who were treated between 1 March 2006 and 31 December 2013 at the Department of Pulmonary Diseases of the Rabin Medical Center, the Israeli National Center for Lung Transplantation. The relevant demographic and clinical data up to December 31, 2015 were retrieved, including post-lung transplantation follow-up status. The diagnosis of silicosis was made according to standard criteria of the US National Institute for Occupational Safety and Health [16] in relation to exposure histories and job tasks consistent with overexposure to silica through the handling of artificial stone. The Israel Ministry of Labour has defined the legal exposure limit in Israel as being 0.1 mg/m^3^ whereupon standard activities, such as dry cutting of the stone, which lead to levels of silica of 1 mg/m^3^, represent a roughly ten-fold level of excess (Israel Ministry of Economy and Industry, 2014).

We applied the data of patients undergoing lung transplantation for IPF at the same medical center during the same time period to serve for comparisons of lung contents of metal particles. None of these patients had any known past occupational history of exposure to any type of hazardous dust. IPF referents were males, restricted to the age range of the silicosis cases but not directly matched for age. Biopsies were prepared from surgical lung transplant explants as previously described [17]. In short, explant samples of ~1.5 cm^3^ were fixed in for 48 h in tissue cassettes and in 10% formalin in 0. 1 M phosphate buffer, pH 7.2, that contained 0.15 M NaCI. Samples were embedded in paraffin into blocks. The whole block (0.5 × 2.5 × 2.5 mm) used for analysis of mineral content was measured with the Niton XL3 XRF analyzer (Thermo Fischer Scientific, Munich, Germany) in the lung biopsies of the IPF patients. Each biopsy was scanned twice by XRF in different areas and an average was calculated. Briefly, the instrument was fitted with an X-ray tube with an Ag anode target excitation source, operating at voltages up to 50 kV and at beam currents up to 200 μA, and a geometrically optimized large area drift detector. A helium purge technique was employed for enhanced light element analysis. A charged coupled device camera stored sample images, and the data were transferred via Thermo Scientific Niton data transfer PC software (Thermo Niton Analyzers LLC, version NDT_REL_8.2.1). Calibration was performed each morning before measurements (RCRApp 1000Ba 500Ag,As,Cd,Pb,Se 180-661; NIST2709a PP 180-649; 180-706pp USGS SdAr-M2 Control Sample; SiO_2_ pp.995%PP 180-647) according with manufacturer instructions. Biopsies were placed in the center of a 6 μm polypropylene film slide over a 3 mm small-spot collimator above the detector. The overall measurement took 300 s, and spectra up to 40 keV were quantified with a factory-installed algorithm (fundamental parameters calibration) for a “mining” mode that yielded elemental concentrations in parts per million (μg g^−1^) with an error of 2*σ* or 95% confidence. A potential source of error can be caused by sample thickness and geometry. In these cases, a known sample thickness less than the geometry limited thickness must be used for accurate intensities. We compared measurements done on a 2 µ slice of paraffin block imposed on a round 25 mm Thermanox 174985 disc to those done on paraffin block. No differences were found (data not shown here)

Pulmonary function tests (PFTs) were performed by means of a Masterlab spirometer (Masterlab E. Jaeger, Wurzburg, Germany). The measurements were carried out according to standard protocols of the American Thoracic Society guidelines [18]. The results are expressed as % predictive value. The best of three consecutive measurements was chosen.

All of the selected ASIS and IPF parameters were compared by independent *t*-test and chi-squared test. Correlations between the levels of all the minerals and the clinical, demographic, and functional parameters were carried out with the Spearman correlation. To determine the effect of age at transplantation (silicosis vs. IPF), we performed univariate Cox regression analyses using these factors as the independent variables. The cut off of silica levels that showed the highest specificity and sensitivity for diagnosing ASIS was calculated with a ROC curve. All statistical analyses were by SPSS (Armonk, NY, USA, IBM, version 22). All *p* values < 0.05 were considered significant.

## 3. Results

The participants’ demographic parameters are described in Table 1. The ASIS patients were significantly younger (48.7 ± 8.8 years vs. 58.2 ± 6.3 years, *p <* 0.001), and they had a lower body mass index at diagnosis and at the time of transplantation compared to the IPF group (24.9 ± 5.2 vs. 28.4 ± 4.3 and 26.3 ± 4.2 and 29.0 ± 4.4, *p <* 0.083 and *p <* 0.044, respectively).

The PFT results are shown in Table 2. The flow expiratory volume in one second (FEV1 percentage of predicted values) were significantly lower among the ASIS patients compared to the IPF group both at diagnosis (51.0 ± 19.1% vs. 63.8 ± 15.0%, respectively, *p <* 0.057) and at the time of transplantation (27.8 ± 9.0% vs. 42.0 ± 16.1%. *p <* 0.004). The ratio between FEV1 and forced vital capacity (FVC) showed a more obstructive pattern in the ASIS group compared to the IPF group (74.5 ± 11.1 vs. 89.0 ± 5.0, *p <* 0.01 at diagnosis time and 71.8 ± 16.5 vs. 83.8 ± 12.4, *p <* 0.025 at transplantation time). The values of diffusion capacity of the lung for carbon monoxide (CO) were not significantly different between groups, but deterioration was clearly observed in both of them.

The chemical analysis in the lung tissues of the study population was carried out by X-ray fluorescence, and it showed values of silica, titanium and aluminum that were around 5- to 10-fold higher in the biopsies of the patients that underwent lung transplantation due to ASIS compared to those who underwent lung transplantation due to IPF (7284.2 ± 4693.7 vs. 898.8 ± 365.6 ppm for silica, *p <* 0.0001; 300.2 ± 151.8 vs. 32.5 ± 39.1 ppm, *p <* 0.0001 for titanium, and 108.6 ± 98.4 vs. 20.4 ± 21.5 ppm, *p <* 0.001 for aluminum, respectively) (Table 3).

The strong correlation between the levels of these metals indicates that the transplanted ASIS workers were exposed to a mixture of alloys containing mainly silica together with titanium, aluminum, zinc and iron (r = 0.911 * < 0.0001 silica titanium and r = 0.890, *p <* 0.0001 silica aluminum) (Table 4). Thresholds of ≥1128 ppm, ≥76.17 ppm and ≥29.42 ppm for silica, titanium and aluminum had high sensitivity and specificity to quantitatively differentiate between the ASIS and the IPF patients.

The FEV1/FVC ratio values were lower for workers with high levels of silica, aluminium, and titanium particles in their lung biopsies, indicating a correlation with obstructive patterns in those patients (Table 5).

## 4. Discussion

We report here the results of analyses of metal contents in the lung biopsies of patients who underwent lung transplantation due to ASIS compared to a group of patients who underwent lung transplantation due to IPF. This is the first analysis using a novel method with quantitative levels of silica and other metals from lung biopsies, and the results showed high specificity and sensitivity to differentiate between specimens of ASIS patients and those of IPF patients.

The standard criteria of the US National Institute for Occupational Safety and Health for the diagnosis of silicosis [16] include exposure histories, industrial hygiene sampling, clinical picture of interstitial lung disease (including radiographic imaging) consistent with silicosis, and mineralogical analysis of dust from the lungs. Multiple mixed exposures are the norm in exposure histories: essentially none of the worksite and environment exposures are to a single substance, and each subject undergoes various exposure combinations during his/her lifetime. Extreme variations of mixed exposures have been reported in certain occupations, such as construction workers, stonecutters and firefighters [19,20]. In this context, exposure history is not always clear for diagnosing silicosis. With regard to industrial hygiene, reports on environmental monitoring are not made by the Israel Ministry of Labour and Economy and the provision of such reports from the small marble workplaces in Israel is haphazard. In the absence of detailed exposure history and the unavailability of regulated reporting of industrial hygiene, mineralogical analysis of dust directly from lung specimens becomes essential for demonstrating disease causation.

Our group had shown that XRF analysis of the induced sputum samples of workers exposed to artificial stone dust revealed that they also contained silica and other minerals [11]. In contrast to the earlier results that measured the metal content in symptomatic and asymptomatic workers as a biomarker of exposure, we measured mineral content directly from ASIS lung biopsies retrieved from workers that underwent lung transplantation due to severe end-stage disease and compared the findings with those derived from biopsies retrieved from individuals that underwent lung transplantation due to IPF. We were able to apply a method of non-destructive elemental analysis to achieve quantitative measurements of metals obtained directly from paraffin tissue blocks of lung explants retrieved from the original silicotic and IPF lungs pre-transplantation.

The SEM is one of the standard methods for imaging the microstructure and morphology and for the identification of metals [21]. In SEM, a low-energy electron beam is radiated to the material and scans the surface of the sample to perform chemical analysis and/or identify the crystalline structure [22]. The results are expressed in a qualitative fashion and the original sample is destroyed in the process. Application of the electron microprobe micro analyser-wavelength dispersive spectroscopy was reported to analyse the content of metals in transbronchial biopsies as well as in liquid specimens (bronchoalveolar lavage and urine) [23,24,25]. However, quantitative analysis of particles in the biopsy specimens of the diseased lungs of patients undergoing lung transplantation due to severe end-stage disease had not been reported.

The patients with silicosis were significantly younger than the IPF patients. Assuming that silica exposure of most workers begins in adulthood, the relatively younger age is consistent with the mean latency of silicosis reported in one study (22.9 years) [26] and with the average age at onset of silicosis of 47.8 years reported in another study [27]. In contrast, IPF presentation typically occurs in the sixth and seventh decades, with a mean age at diagnosis of 66 years [28].

Although the FEV1 of the ASIS patients was significantly lower than that of the IPF patients, the post-lung transplantation survival was reportedly similar for both groups [29]. The mixed restrictive and obstructive pattern observed in the current study is common in silicosis, particularly in the presence of progressive massive fibrosis [30].

The silica, titanium, aluminum iron and zinc contents were significantly higher in the tissues retrieved from the lungs of the ASIS patients compared to those of the IPF patients. Moreover, there was a strong correlation between the levels of those metals, alluding to the composition of the final product of pigmented artificial stone that was already shown by us as containing many metals [11].

The first composition formula of artificial stone was patented in 1889, and it already documented the presence of sand and metals, such as aluminum and zinc [31]. Another patent of artificial stone refers to a mixture of inorganic components together with resins [32]. The composition of the Israeli artificial stone dust had not been described in the Safety Data Sheet of the product until 2019 [33]. It was reported to contain <93% silica, <4% titanium, and other “pigments”, and included a warning that inhalation of the dust product can cause silicosis. The first cases of ASIS, however, were described in 2006 [3], suggesting that workers had long been exposed to artificial stone dust without any knowledge about the correlation and causation between it and ASIS.

Our results showed significantly higher silica, titanium, aluminium and zinc content in the tissues of the ASIS tissue biopsies compared to those contents in the IPF tissue biopsies. Most interesting is the presence of titanium [34,35] and aluminum [33], given that these metals are known to cause granulomatous diseases [36]. In this context, we already showed an outbreak of autoimmune diseases in workers exposed to artificial stone dust, with 6/9 cases having also shown mediastinal lymphadenopathy compatible with granulomatous diseases [37]. In a recently reported paper, it was shown that multiorgan accelerated silicosis was misdiagnosed as sarcoidosis in two workers exposed to quartz conglomerate dust [38].

Qualitative analysis of metal content in tissue or induced sputum specimens makes it possible to rule out the idiopathic element of interstitial lung disease of occupationally exposed workers and points to the causative agent(s) [10]. IPF has been well defined by precise criteria for the radiological and histopathological features of interstitial pneumonia [29]. Quantitative analysis with XRF spectrometry, however, revealed a clear cut off for silica levels with high specificity and sensitivity in differentiating between ASIS and IPF. The diagnosis of ASIS can be facilitated by combining XRF spectrometry findings with other clinical and roentgenologic parameters where environmental/occupational anamnesis is neglected.

The main limitation of the present study is being a pioneer in the use of XRF for the analysis of metal content in biological samples. Further studies with larger population will be valuable to standardize this rapid, simple and none-destructive method to quantitate metals in the lung.

## 5. Conclusions

In conclusion, many workers change their workplaces and environmental settings during their lifetime, and the length of exposure to any noxious inhalant is not always clear. The provision of a cut off of silica content in the lung tissue may be pivotal for a definitive diagnosis of serious pathology and for the claim of compensation for artificial stone-induced lung injury.

## Figures and Tables

**Table 1 ijerph-18-07237-t001:** Demographic and clinical parameters of the study population (34 patients).

Variables	ASIS	IPF	*p*-Value *
Age at diagnosis, y	48.7 ± 8.8	58.2 ± 6.3	0.001
BMI (at diagnosis)	26.3 ± 4.2	29.0 ± 4.4	0.083
BMI (at transplantation)	24.9 ± 5.2	28.4 ± 4.3	0.044
Pack years (smoking)	26.8 ± 15.8	32.5 ± 13.0	0.392

Data are presented as mean ± SD. ASIS, artificial stone-induced silicosis; IPF, idiopathic pulmonary fibrosis; BMI, body mass index. * *p <* 0.05 for age, body mass index, silica exposure, smoking (independent *t*-test), and survival (chi-squared test).

**Table 2 ijerph-18-07237-t002:** Pulmonary function tests at diagnosis of the disease and at the time of lung transplantation.

Variables	At Diagnosis			At Transplantation		
% Predicted	ASIS	IPF	*p*-Value	ASIS	IPF	*p*-Value
FVC	50.0 ± 14.0 n = 12	59.3 ± 12.8 n = 14	0.321	32.3 ± 8.7 n = 17	39.3 ± 15.1 n = 16	0.113
FEV1	51.0 ± 19.1 n = 13	63.8 ± 15.0 n = 15	0.057	27.8 ± 9.0 n = 17	42.0 ± 16.1 n = 16	0.004
FEV1/FVC	74.5 ± 11.1 n = 10	89.0 ± 5.0 n = 12	0.01	71.8 ± 16.5 n = 17	83.8 ± 12.4 n = 16	0.025
TLC.	59.4 ± 11.8 n = 11	65.5 ± 12.4 n = 13	0.235	50.7 ± 14.0 n = 16	47.5 ± 12.3 n = 12	0.541
DLCO	47.4 ± 14.8 n = 12	51.4 ± 18.4 n = 12	0.565	34.0 ± 12.0 n = 16	31.3 ± 11.5 n = 11	0.574

ASIS, artificial stone-induced silicosis; IPF, idiopathic pulmonary fibrosis; FVC, forced vital capacity; FEV1, forced expiratory volume in 1 s; TLC, total lung capacity; DLCO sb, single-breath diffusing capacity of the lung for carbon monoxide; VA, alveolar volume; *p <* 0.05 independent *t*-test.

**Table 3 ijerph-18-07237-t003:** Chemical analysis by x-ray fluorescence in lung tissues (34 patients).

Variables	ASIS	IPF	*p*-Value *
Silica	7284.2 ± 4693.7	898.8 ± 365.6	<0.0001
Iron	1126.2 ± 1573.9	418.2 ± 636.1	0.095
Titanium	300.2 ± 151.8	32.5 ± 39.1	<0.0001
Aluminum	108.6 ± 98.4	20.4 ± 21.5	0.001
Zinc	13.8 ± 10.1	23.1 ± 9.9	0.001

Data are presented as the mean ± SD of four measurements and expressed in parts per million (ppm); ASIS, artificial stone-induced silicosis; IPF, idiopathic pulmonary fibrosis. * *p <* 0.05 (independent *t*-test).

**Table 4 ijerph-18-07237-t004:** Correlation between silica and other mineral contents in lung tissues (34 patients).

Variables	Silica	
	r	*p*-value
Titanium	0.911 *	<0.0001
Sulfur	0.145	0.412
Iron	0.525 *	0.001
Zinc	0.340 †	0.049
Aluminium	0.890 *	<0.0001

* *p <* 0.01 (Spearman correlation); † *p <* 0.05 (Spearman correlation).

**Table 5 ijerph-18-07237-t005:** Correlation between mineral content and pulmonary function tests (34 patients).

Variables	Silica	Titanium	Aluminium
FEV1/FVC at diagnosis	−0.748 *	−0.637 *	−0.686 *
FEV1/FVC at transplantation	−0.193	−0.344 *	−0.060

* *p <* 0.05 Spearman correlation; FEV1, forced expiratory volume in 1 s; FVC, forced vital capacity.

## Data Availability

Data is available in patients records at Institute of Pulmonary Diseases, Rabin Medical Centre.

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
