# Peer review of "Quantitation of Silica Contents in Lung Explants of Transplanted Patients: Artificial Stone-Induced Silicosis vs. Idiopathic Pulmonary Fibrosis"

_ijerph, 2021, doi:10.3390/ijerph18147237_

Round 1

Reviewer 1 Report

Reviewer

Abstract

Comment:

It is suitable

Introduction

      Lines - 47- 49 - We first used SEM to redefine idiopathic interstitial lung disease into occupational lung diseases by analyzing the chemical composition of inhaled dust particles in induced sputum and/or lung biopsy specimens    10  ?

Comment:

This paragraph could be written:.. in our previous work, we determined that ......

  1. Fireman E, Lerman Y, Ben Mahor M, et al. Redefining idiopathic interstitial lung disease into occupa-282 tional lung diseases by analysis of chemical composition of inhaled dust particles in induced sputum 283 and/or lung biopsy specimens. Toxicol Ind Health 2007; 23: 607–615.

Line- 53- ……for what we believe to be the first time….

Comment:

Wouldn't it be interesting to put these comments in the discussion?

Materials and Methods

Lines - 62- 65-  We conducted a retrospective review of the medical records of 17 ASIS patients and  17 IPF patients who were treated between March 1, 2006 and December 31, 2013 at the  Department of Pulmonary Diseases of the Rabin Medical Center, the Israeli National 64 Center for Lung Transplantation 12 .

  1. Kramer MR, Blanc PD, Fireman E, et al. Artificial stone silicosis [corrected]: Disease resurgence among 287 artificial stone workers. Chest 2012;142 (2):419–424.

Comment:

How many were men and how many were women?

It is the age?

Statistical analyses

It is suitable

Results

Lines 109-110 - The participants’ demographic parameters are described in Table 1. The ASIS patients were significantly younger (48.7 ± 8.8 years vs 58.2 ± 6.3 years ….

Comment:

The age of patients in Materials and Methods was missing.

Discussion

Lines - 165 - 168- Multiple mixed exposures are the norm in exposure histories: essentially none of worksite and environment  exposures are to a single substance, and each subject undergoes various exposure combinations during his/her lifetime.

Comment:

In Materials and Methods, sex was lacking.

Lines - 198-203 - The patients with silicosis were significantly younger than the IPF patients. Assuming that silica exposure of most workers begins in adulthood, the relatively younger age is consistent with the mean latency of silicosis reported in one study (22.9 years) 23 and  with the average age at onset of silicosis of 47.8 years reported in another study 24. In contrast, IPF presentation typically occurs in the sixth and seventh decades, with a mean age at diagnosis of 66 years 25.

Comment:

According to this sentence it is important to put in the Materials and Methods the age of the 34 patients.

Conclusion

Comment:

It is suitable

References

Line - 270 -     6. Bartoli D, Banchi B, Di Benedetto F, et al. Silicosis….

Comment:

Why are the names of the authors in this reference in bold?

Lines 277- 281 -    8. Lucinda B, Frankel A. Yates D. Complicated silicosis in an Australian worker from cutting engineered stone countertops. An embarrassing first for Australia. Abstract presented ERS Conference Amsterdam 24-30, 2015.

  1. Abrahams JL, Analysis of Fibrous and non Fibrous Particles in Environmental and Occupational in Medicine NW Rom Markovitz SB 2007 4Th edition.

Comment:

Why are these references written in bold?

Lines- 285-286 -        11. Ophir N, Shai AB, Alkalay Y, et al. Artificial stone dust-induced functional and inflammatory abnor-285 malities in exposed workers monitored quantitatively by biometrics. ERJ Open Res. 2016 Mar 17;2(1)

Comment:

Why is the name of the article written in blue in this reference?

Lines - 289- 295-      13. Centers for Disease Control (CDC), National Institute for Occupational Safety and Health  (NIOSH). 2002. Health effects of occupational exposure to respirable crystalline silica. Washington, DC: National Institute for Occupational Safety and Health. p.129. http://www.cdcgov/niosh/docs/2002-129/pdfs/2002-129.pdf Accessed: December 7,2014. 

  1. Berry GJ. Lung Transplantation in Pathology of Transplantation. Springer Int 2016: 121-172.

  1. Miller MR, Crapo R, Hankinson J, Brusasco Vet al. General considerations for lung function testing. ERJ 2005. 26:153–161.

Comment:

Why are these references written in bold?

  1. Line - 296- https://www.osha.gov/Publications/silicosis.html

Comment:

Shouldn't this reference be written in blue?

Lines- 299-301-      18. Ophir N, Bar Shai, Alkalay Y, et al. Artificial stone dust-induced functional and inflammatory abnormalities in exposed workers monitored quantitatively by biometrics. ERJ Open Res. 2016 Mar 17;2(1).

Comment:

Why is this reference written in bold?

Lines- 310-312-     22. Hiromi T, Toshihiko K, Eiji K, et al. Elemental analysis of occupational granulomatous lung disease by electron probe microanalyzer with wavelength dispersive spectrometer: Two case reports. Respir Med  Case Reports 2016; 18:66-72. 

Comment:

Why are the authors' names written in blue and also the name of the journal?

Lines- 317-319-       25. Raghu G, Collard HR, Egan JJ, et al. An Official ATS/ERS/JRS/ALAT Statement: Idiopathic Pulmo-317 nary Fibrosis: Evidence-based Guidelines for Diagnosis and Management Am J Respir Crit Care Med 318 2011; 183(6): 788–824

Comment:

Why are the authors' names written in blue in this reference?

Lines- 329- 330 -      32. Redline S, Barna BP, Tomashefski JF Jr, Abraham JL. Granulomatous disease associated with pulmo-329 nary deposition of titanium. Br J Ind Med. 1986 Oct; 43(10):652-6.

Comment:

Why is the article name written in blue?

Lines- 331-332-    33. Chen WJ, Monnat RJ Jr, Chen M, Mottet NK. Aluminum Induced Pulmonary Granulomatosis Hum Pathol 1978 Nov;9(6):705-11

Comment:

Why are the authors' names written in blue?

Thank you

Author Response

Answer to reviewer is attached

Reviewer 2 Report

Major - More references needed demonstrating the use of XRF to quantitatively determine mineral content in biological samples would be beneficial.

Minor - Any references that compare XRF to other analytical techniques such as ICP-MS or others for minerals in the ppm range of biological samples would be helpful.

Major - Expansion of the XRF explanation in the methods section and/or discussion section will help the reader understand the potential error in this quantitation technique.

Lines 47-49 requires font correction.

Author Response

The reply to reviewer is attached

Reviewer 3 Report

X-ray fluorescence is an elemental method of analysis and cannot determine minerals compounds directly.  The algorithm of the handheld instrument is matrix dependent and should be validated using known samples in a similar matrix as the unknown samples.  Further, the instrument used in this study is not capable of detecting oxygen, so any "quantitation" of silica is primarily based off of detection of silicon.  As other silicate minerals may be present (for instance, detection of both silicon and aluminum could suggest an aluminosilicate), it is not appropriate to attribute all silicon to silica unless a complementary analysis such as XRD is carried out.

I do feel there is value in this work as it attempts to address a gap in the current body of knowledge, but at a minimum a more detailed discussion of the limitations of the XRF method, and a more careful interpretation of results, is necessary.  Additional laboratory analysis would also be appropriate.

Author Response

The report for reviewer is attached

Reviewer 4 Report

this is a very interesting manuscript by Dr. Elizabeth Fireman and colleagues who describe the method of silica quantification in lung biopsy specimens as diagnostic aid for silicosis in workers of artificial stone.

Page 2, line 79:  The type of lung biopsy specimen used for the silica analysis has to be explained in more detail, just a reference is not sufficient.  This part is critical for attempts to reproduce these findings.  For example, what does it mean 'paraffin embedded tissue ' mean?  Were the lungs formaldehyde fixed (concentration / for how long, perhaps there are ranges for both, e.g. 4-10%, 2 days - years).  Or was the lung tissue snap frozen.  Then paraffin tissue blocks were made & H&E stained sections obtained with evidence for silica induced lung disease on light microscopy ?  Were further slices cut and pieces cut based on the location of the previously identified nodules ?  Or was the whole remaining block of tissue used for analysis - if yes, how much of the remaining whole block - what was the volume of tissue used for analysis:  um thickness x block size (mm x mm).

Page 2: line 81:  metal content measurements - were tests implemented to confirm the accuracy of the instrument ?  e.g. standard curves with known quantities of the metals to be tested, alone and in pre-mixed, standardized concentrations ?  This is important because the Niton XL3 XRF analyzer (Thermo Fischer 81 Scientific, Munich, Germany) seems to be an industrial testing instrument for field analysis.

Results:  The tables clearly show the data, but it would be nice to convert for example table 3 or table 5 into a figure (with several panels) that also show individual data points.  This would be easier to read than a table, and would also show overlap of data between groups & the correlations with lung function data.

Author Response

Report for reviewer is attached

Round 2

Reviewer 4 Report

The method details were in part updated and addressed.  However, the issue with instrument calibration was not addressed, only in a sentence in the response to reviewers section.  I think for research transparency, it is important to address the standardization, the readers need to know - was the instrument standardized, how frequently was standardization performed, and which method was used, e.g. calibration kit provided by the manufacturer.

The same applies to the actual metal measurements. The sentences (line 100 onwards) are very helpful because they describe an important aspect of the method used that is essential for repeatability and transparency of the research.  However, there needs to be a follow-on sentence to this: "In those case a known sample thickness less than the geometry limited thickness must be used for accurate intensities."  For research transparency, the reader would need to know how the variables (known sample thickness, and geometry limited thickness) were determined.   

Author Response

The method details were in part updated and addressed.  However, the issue with instrument calibration was not addressed, only in a sentence in the response to reviewers section.  I think for research transparency, it is important to address the standardization, the readers need to know - was the instrument standardized, how frequently was standardization performed, and which method was used, e.g. calibration kit provided by the manufacturer.

The same applies to the actual metal measurements. The sentences (line 100 onwards) are very helpful because they describe an important aspect of the method used that is essential for repeatability and transparency of the research.  However, there needs to be a follow-on sentence to this: "In those case a known sample thickness less than the geometry limited thickness must be used for accurate intensities."  For research transparency, the reader would need to know how the variables (known sample thickness, and geometry limited thickness) were determined.

Calibration was performed each morning before measurements (RCRApp 1000Ba 500Ag,As,Cd,Pb,Se 180-661; NIST2709a PP 180-649; 180-706pp USGS SdAr-M2 Control Sample; SiO2 pp.995%PP 180-647) according with manufacturer instructions.  Biopsies were placed in the center of a 6 μm polypropylene film slide over a 3 mm small-spot collimator above the detector. The overall measurement took 300 seconds, and spectra up to 40 keV were quantified with a factory-installed algorithm (fundamental parameters calibration) for a “mining” mode that yielded elemental concentrations in parts per million (μg g−1) with an error of 2σ or 95% confidence. A potential source of error can be caused by sample thickness and geometry. In those case a known sample thickness less than the geometry limited thickness must be used for accurate intensities. We compare measurements done on a 2µ slice of paraffin block imposed on a round 25 mm Thermanox 174985 disc to those done on paraffin block. No difference were found (data not shown here)

This appear now in rows 95-108

The method details were in part updated and addressed.  However, the issue with instrument calibration was not addressed, only in a sentence in the response to reviewers section.  I think for research transparency, it is important to address the standardization, the readers need to know - was the instrument standardized, how frequently was standardization performed, and which method was used, e.g. calibration kit provided by the manufacturer.

The same applies to the actual metal measurements. The sentences (line 100 onwards) are very helpful because they describe an important aspect of the method used that is essential for repeatability and transparency of the research.  However, there needs to be a follow-on sentence to this: "In those case a known sample thickness less than the geometry limited thickness must be used for accurate intensities."  For research transparency, the reader would need to know how the variables (known sample thickness, and geometry limited thickness) were determined.

Calibration was performed each morning before measurements (RCRApp 1000Ba 500Ag,As,Cd,Pb,Se 180-661; NIST2709a PP 180-649; 180-706pp USGS SdAr-M2 Control Sample; SiO2 pp.995%PP 180-647) according with manufacturer instructions.  Biopsies were placed in the center of a 6 μm polypropylene film slide over a 3 mm small-spot collimator above the detector. The overall measurement took 300 seconds, and spectra up to 40 keV were quantified with a factory-installed algorithm (fundamental parameters calibration) for a “mining” mode that yielded elemental concentrations in parts per million (μg g−1) with an error of 2σ or 95% confidence. A potential source of error can be caused by sample thickness and geometry. In those case a known sample thickness less than the geometry limited thickness must be used for accurate intensities. We compare measurements done on a 2µ slice of paraffin block imposed on a round 25 mm Thermanox 174985 disc to those done on paraffin block. No difference were found (data not shown here)

This appear now in rows 95-108

The method details were in part updated and addressed.  However, the issue with instrument calibration was not addressed, only in a sentence in the response to reviewers section.  I think for research transparency, it is important to address the standardization, the readers need to know - was the instrument standardized, how frequently was standardization performed, and which method was used, e.g. calibration kit provided by the manufacturer.

The same applies to the actual metal measurements. The sentences (line 100 onwards) are very helpful because they describe an important aspect of the method used that is essential for repeatability and transparency of the research.  However, there needs to be a follow-on sentence to this: "In those case a known sample thickness less than the geometry limited thickness must be used for accurate intensities."  For research transparency, the reader would need to know how the variables (known sample thickness, and geometry limited thickness) were determined.

Calibration was performed each morning before measurements (RCRApp 1000Ba 500Ag,As,Cd,Pb,Se 180-661; NIST2709a PP 180-649; 180-706pp USGS SdAr-M2 Control Sample; SiO2 pp.995%PP 180-647) according with manufacturer instructions.  Biopsies were placed in the center of a 6 μm polypropylene film slide over a 3 mm small-spot collimator above the detector. The overall measurement took 300 seconds, and spectra up to 40 keV were quantified with a factory-installed algorithm (fundamental parameters calibration) for a “mining” mode that yielded elemental concentrations in parts per million (μg g−1) with an error of 2σ or 95% confidence. A potential source of error can be caused by sample thickness and geometry. In those case a known sample thickness less than the geometry limited thickness must be used for accurate intensities. We compare measurements done on a 2µ slice of paraffin block imposed on a round 25 mm Thermanox 174985 disc to those done on paraffin block. No difference were found (data not shown here)

This appear now in rows 95-108

The method details were in part updated and addressed.  However, the issue with instrument calibration was not addressed, only in a sentence in the response to reviewers section.  I think for research transparency, it is important to address the standardization, the readers need to know - was the instrument standardized, how frequently was standardization performed, and which method was used, e.g. calibration kit provided by the manufacturer.

The same applies to the actual metal measurements. The sentences (line 100 onwards) are very helpful because they describe an important aspect of the method used that is essential for repeatability and transparency of the research.  However, there needs to be a follow-on sentence to this: "In those case a known sample thickness less than the geometry limited thickness must be used for accurate intensities."  For research transparency, the reader would need to know how the variables (known sample thickness, and geometry limited thickness) were determined.

Calibration was performed each morning before measurements (RCRApp 1000Ba 500Ag,As,Cd,Pb,Se 180-661; NIST2709a PP 180-649; 180-706pp USGS SdAr-M2 Control Sample; SiO2 pp.995%PP 180-647) according with manufacturer instructions.  Biopsies were placed in the center of a 6 μm polypropylene film slide over a 3 mm small-spot collimator above the detector. The overall measurement took 300 seconds, and spectra up to 40 keV were quantified with a factory-installed algorithm (fundamental parameters calibration) for a “mining” mode that yielded elemental concentrations in parts per million (μg g−1) with an error of 2σ or 95% confidence. A potential source of error can be caused by sample thickness and geometry. In those case a known sample thickness less than the geometry limited thickness must be used for accurate intensities. We compare measurements done on a 2µ slice of paraffin block imposed on a round 25 mm Thermanox 174985 disc to those done on paraffin block. No difference were found (data not shown here)

This appear now in rows 95-108

The method details were in part updated and addressed.  However, the issue with instrument calibration was not addressed, only in a sentence in the response to reviewers section.  I think for research transparency, it is important to address the standardization, the readers need to know - was the instrument standardized, how frequently was standardization performed, and which method was used, e.g. calibration kit provided by the manufacturer.

The same applies to the actual metal measurements. The sentences (line 100 onwards) are very helpful because they describe an important aspect of the method used that is essential for repeatability and transparency of the research.  However, there needs to be a follow-on sentence to this: "In those case a known sample thickness less than the geometry limited thickness must be used for accurate intensities."  For research transparency, the reader would need to know how the variables (known sample thickness, and geometry limited thickness) were determined.

Calibration was performed each morning before measurements (RCRApp 1000Ba 500Ag,As,Cd,Pb,Se 180-661; NIST2709a PP 180-649; 180-706pp USGS SdAr-M2 Control Sample; SiO2 pp.995%PP 180-647) according with manufacturer instructions.  Biopsies were placed in the center of a 6 μm polypropylene film slide over a 3 mm small-spot collimator above the detector. The overall measurement took 300 seconds, and spectra up to 40 keV were quantified with a factory-installed algorithm (fundamental parameters calibration) for a “mining” mode that yielded elemental concentrations in parts per million (μg g−1) with an error of 2σ or 95% confidence. A potential source of error can be caused by sample thickness and geometry. In those case a known sample thickness less than the geometry limited thickness must be used for accurate intensities. We compare measurements done on a 2µ slice of paraffin block imposed on a round 25 mm Thermanox 174985 disc to those done on paraffin block. No difference were found (data not shown here)

This appear now in rows 95-108

The method details were in part updated and addressed.  However, the issue with instrument calibration was not addressed, only in a sentence in the response to reviewers section.  I think for research transparency, it is important to address the standardization, the readers need to know - was the instrument standardized, how frequently was standardization performed, and which method was used, e.g. calibration kit provided by the manufacturer.

The same applies to the actual metal measurements. The sentences (line 100 onwards) are very helpful because they describe an important aspect of the method used that is essential for repeatability and transparency of the research.  However, there needs to be a follow-on sentence to this: "In those case a known sample thickness less than the geometry limited thickness must be used for accurate intensities."  For research transparency, the reader would need to know how the variables (known sample thickness, and geometry limited thickness) were determined.

Calibration was performed each morning before measurements (RCRApp 1000Ba 500Ag,As,Cd,Pb,Se 180-661; NIST2709a PP 180-649; 180-706pp USGS SdAr-M2 Control Sample; SiO2 pp.995%PP 180-647) according with manufacturer instructions.  Biopsies were placed in the center of a 6 μm polypropylene film slide over a 3 mm small-spot collimator above the detector. The overall measurement took 300 seconds, and spectra up to 40 keV were quantified with a factory-installed algorithm (fundamental parameters calibration) for a “mining” mode that yielded elemental concentrations in parts per million (μg g−1) with an error of 2σ or 95% confidence. A potential source of error can be caused by sample thickness and geometry. In those case a known sample thickness less than the geometry limited thickness must be used for accurate intensities. We compare measurements done on a 2µ slice of paraffin block imposed on a round 25 mm Thermanox 174985 disc to those done on paraffin block. No difference were found (data not shown here)

This appear now in rows 95-108

The method details were in part updated and addressed.  However, the issue with instrument calibration was not addressed, only in a sentence in the response to reviewers section.  I think for research transparency, it is important to address the standardization, the readers need to know - was the instrument standardized, how frequently was standardization performed, and which method was used, e.g. calibration kit provided by the manufacturer.

The same applies to the actual metal measurements. The sentences (line 100 onwards) are very helpful because they describe an important aspect of the method used that is essential for repeatability and transparency of the research.  However, there needs to be a follow-on sentence to this: "In those case a known sample thickness less than the geometry limited thickness must be used for accurate intensities."  For research transparency, the reader would need to know how the variables (known sample thickness, and geometry limited thickness) were determined.

Calibration was performed each morning before measurements (RCRApp 1000Ba 500Ag,As,Cd,Pb,Se 180-661; NIST2709a PP 180-649; 180-706pp USGS SdAr-M2 Control Sample; SiO2 pp.995%PP 180-647) according with manufacturer instructions.  Biopsies were placed in the center of a 6 μm polypropylene film slide over a 3 mm small-spot collimator above the detector. The overall measurement took 300 seconds, and spectra up to 40 keV were quantified with a factory-installed algorithm (fundamental parameters calibration) for a “mining” mode that yielded elemental concentrations in parts per million (μg g−1) with an error of 2σ or 95% confidence. A potential source of error can be caused by sample thickness and geometry. In those case a known sample thickness less than the geometry limited thickness must be used for accurate intensities. We compare measurements done on a 2µ slice of paraffin block imposed on a round 25 mm Thermanox 174985 disc to those done on paraffin block. No difference were found (data not shown here)

This appear now in rows 95-108
